**Data Availability Statement:** Data collected in this study were from a small number of participants and the focus group discussions contain potentially

# "I would watch her with awe as she swallowed the first handful": A qualitative study of pediatric multidrug-resistant tuberculosis experiences in Durban, South Africa

Shriya Misra[1], Nirupa Misra[2], Boitumelo Seepamore[3], Kerry Holloway[2], Nalini Singh[2], Jacqui Ngozo[4], Vusi Dlamini[5], Zanele Radebe[4], Norbert O. Ndjeka[6], Jennifer Furin[7,8]*

1 The Health Ninja, Durban, South Africa, 2 King Dinuzulu Hospital Complex, Centralized Drug Resistant Tuberculosis Hospital, Durban, South Africa, 3 University of KwaZulu-Natal, Durban, South Africa, 4 KwaZulu-Natal Provincial HAST Unit, Durban, South Africa, 5 KwaZulu-Natal, Provincial Department of Health, Durban, South Africa, 6 National Department of Health, National TB Control and Management Cluster Office, Pretoria, South Africa, 7 Sentinel Project on Pediatric Drug Resistant Tuberculosis, Boston, Massachusetts, United States of America, 8 Department of Global Health and Social Medicine, Harvard Medical School, Boston, Massachusetts, United States of America

* jenniferfurin@gmail.com

## Abstract

### Background

There are limited data on the experiences of children being treated for drug-resistant tuberculosis (DR-TB), and most work in the area has been done with older children and adolescents. Comprehensive explorations of the caregiver experiences in this area are also lacking.

### Objective

To describe the experiences of being treated for drug-resistant tuberculosis of children and their caregivers.

### Methods

This was a qualitative study done using focus group discussions (FGDs) among three different groups of participants: 1) health care providers involved in the care of children being treated for DR-TB (including physicians, nurses, and pharmacists)—herein referred to as *providers*; 2) household caregivers of children being treated for DR-TB—herein referred to as *caregivers*; and 3) children who were being treated for DR-TB—herein referred to as *children*. The population was a convenience sample and included children hospitalized between January 1, 2018, and June 30, 2020, ages 0–14 years old, as well as their caregivers and providers. Focus group transcripts and notes were analysed using a thematic network analysis based in grounded theory The analysis was iterative and the coding system developed focused on "stressful experiences" as well as ways to address them along the diagnostic and treatment journey. This paper follows the COREQ guidelines.

identifiable data. Thus, the ethics review board did not approve making the data widely available on open access. Thus there are ethical limitations to providing the data. Our ethics review board did not approve this being open access since it could lead to identification of participants. We have provided our interview guides and coding framework. If there are any requests to view the data, they can be sent to the UMgungundlovu Health Ethics Research Board in Durban, South Africa by contacting the chairperson and administrator of the ERB at: Chairperson: Dr Damian Clarke Email: damianclar@gmail.com; Administrator: Nqobile Makhathini (033 395 2102) Email: nqobile. makhathini@kznhealth.gov.za.

**Funding:** This work was supported by a grant from the Stop TB Partnership's Global Drug Facility. The funders had no role in study design, data collection and analysis, decision to publish, or preparation of the manuscript.

**Competing interests:** The authors have declared that no competing interests exist.

## Results

16 children between the ages 7 and 14 years participated in 5 FGDs, 30 caregivers participated in 7 FGDs, and 12 providers participated in 3 FDGs. Data from the children and the caregivers were the focus of this analysis, although some themes were informed by the discussions with the providers as well. In general, it was reported that for a child diagnosed with DR-TB, there is a lived experience of stress that impacts their physical, mental, and social well-being. These pediatric patients and their families therefore develop strategies for coping with these disruptions to their lives. In general, there were major disruptive experiences that resulted from the process around receiving a *diagnosis* of DR-TB and second distinct set of stressful experiences that occurred during the *treatment* of DR-TB once the diagnosis had been made. These stresses occur in the physical, mental, and social realms, and families develop multiple strategies to cope with them, demonstrating resilience in the face of this disease.

## Conclusion

Addressing the stresses experienced by children and their caregivers through child-friendly DR-TB testing, treatment, and counseling is not only essential for ending TB but also for enacting a human-rights based approach to child health in general. Children with DR-TB are a vulnerable population, and they have often been the last to benefit from advances in general pediatric care and in DR-TB care more specifically.

## Background

Children are a vulnerable population when it comes to tuberculosis (TB) and drug-resistant tuberculosis (DR-TB) [1]. Lack of access to diagnosis is one of the most serious challenges in the field of pediatric DR-TB: globally, it is estimated that fewer than 20% of the children who develop DR-TB each year are ever diagnosed with the disease [2]. Even in high-burden countries with strong pediatric DR-TB programs—such as South Africa—pediatric DR-TB is under-diagnosed, representing less than 5% of the adult burden reported to National TB Programs [3]. Although under-diagnosis remains the serious challenge when it comes to DR-TB in children, pediatric DR-TB treatment itself is also fraught with difficulty, including prolonged hospitalizations, lack of access to novel therapeutics, and the use of toxic medications that are difficult to take and tolerate [4]. Despite higher rates of treatment success reported in children compared to adults with DR-TB [5], their therapeutic experiences may be highly stressful [6].

While there is a growing literature on the treatment journeys of adults and adolescents living with DR-TB [7], little is known about the experiences of children who are diagnosed and treated for the disease. This means that there are almost no data in this population that could be used to develop more "child-centered" approaches to DR-TB treatment [8]. This is especially problematic since younger children often require tailored approaches to their care that can address their unique developmental stages; may need pediatric formulations of medications; and are dependent on family members and other adults for successful treatment completion [9]. In order to address this gap in our understanding of optimal pediatric DR-TB care, we undertook a qualitative study of the diagnostic and therapeutic experiences of younger

children, receiving care for DR-TB in Durban, South Africa and their caregivers. Providers of healthcare services for these children were also interviewed to validate what was reported by the children and their caregivers.

## Methods

### Study design

This was a qualitative study done using focus group discussions (FGDs) among three different groups of participants: 1) health care providers involved in the care of children being treated for DR-TB (including physicians, nurses, and pharmacists) but who were not responsible for the care of the children outside of the health care setting—herein referred to as *providers*; 2) household caregivers of children being treated for DR-TB who were responsible for looking after the child in his/her household and who could be parents, other relatives, or legal guardians—herein referred to as *caregivers*; and 3) children who were being treated for DR-TB— herein referred to as *children*. The population was a convenience sample and included children hospitalized between January 1, 2018, and June 30, 2020, ages 7–14 years old, as well as the caregivers and providers of hospitalized children of all ages. Of note, for this study, children were defined as those between 0 and 14 years because this is how the pediatric population has historically been defined in the provision of TB services by both the World Health Organization and the South African National Department of Health.

### Study setting and population

The children, caregivers and providers who participated in this study came from a single centralised DR-TB hospital that treats children and adults with DR-TB in Durban, KwaZulu-Natal, South Africa. All children in the province of KwaZulu-Natal who are diagnosed with DR-TB (either bacteriologically confirmed or clinically diagnosed) are referred to this hospital to start treatment according to South African guidelines. They are hospitalized for a period of time, depending on the clinical status and evolution, then discharged home to complete their treatment. Children between the ages of 7 and 14 years and the caregivers of children of all ages who were hospitalized and their healthcare providers were eligible. Children were enrolled in the study after obtaining consent from their parent/guardian and formal assent from the child if the child was age 12 years or older (according to South African regulations). A total of 42 children were eligible to participate in the study, 42 of their parents/guardians were contacted to see if they would allow the child to participate, 31 agreed telephonically to allow the child to participate, however only 16 participated in the study. Reasons given by parents/guardians for child non-participation included that they felt the child was too young to discuss his/her treatment journey or that the treatment was too long ago for the child to remember. For each child who participated, at least one caregiver was asked to participate in a separate FGD targeted at adult caregivers. A total of 143 caregivers of children were eligible to participate, 92 were able to be contacted and asked to participate, 66 agreed telephonically and 30 arrived and participated. Reasons for non-participation included travel distance to hospital, work commitments that did not allow for participation, and the length of time that had passed since the child was treated. In terms of providers, any health care professional who participated in the care of a child with DR-TB during the study period was considered eligible to participate. A total of 16 providers were eligible to participate in the study, 16 were asked to participate, and 12 agreed and participated. One healthcare worker was on leave and 3 were not available to participate.

## Data collection and analysis

A total of 16 children between the ages 7 and 14 years participated in 5 FGDs (with between 3 and 4 children per group), 30 caregivers of children ages 0 to 14 years participated in 7 FGDs (with between 4 and 5 adults per group), and 12 providers participated in 3 FGDs (with 4 providers per group). All focus groups contained the same category of participants (i.e., caregiver focus groups contained only caregivers). The caregivers' and children's FGD groups were also categorized according to the age groups of the children (0 to 6 years for caregivers only; 7–10 years; and 10 to 14 years for children and caregivers, although no children between the ages of 0 to 7 years themselves participated in the discussions due to their ages making participation difficult).

After establishing a rapport, focus group discussions were led by two female facilitators (BS and SM with PhD. And M.A degrees respectively: BS has multiple years of experience leading such focus groups). An interview guide was designed and used, to ask them about the experiences of their DR-TB treatment journey, the medicine that had to be administered to the children, the difficulties experienced throughout their treatment, and their recommendations for children and caregivers that will go through this journey in the future, which can be seen in the S1 File. All focus groups were conducted by well-trained individuals, in both English and in isiZulu (depending on the group's preference) and recorded. The recordings were then transcribed (by persons external to the research team but who are trained and certified in ethical research practice as well as the ethical handling of sensitive information) and then translated into English, where necessary for data analysis. Participants were told about the reason for invitation to the study, and who the interviewers were. Each participant was only in one FGD, and the focus group interviews lasted 30 and 90 minutes. Field notes were not kept or analyzed. Transcripts were not reviewed by the participants.

Data analysis was based in grounded theory, which centers the analysis on the accounts of the study participants, as opposed to using an already-existing analytic framework [10]. Grounded theory was selected as little is known about the experiences of children who are undergoing treatment for DR-TB. The analysis was, however, informed by a trauma framework developed by Das and colleagues in their study on the illness experiences of adolescents with DR-TB living in Mumbai, India [4] which described the treatment experiences these adolescents had as painful, damaging to their physical wellbeing and sense of self, and disrupting normal social roles and development activities "Trauma" is a term sometimes used to describe the experiences of children with chronic and/or life-threatening illness [11]. Trauma may be defined "as physical and psychological experiences that are distressing, emotionally painful, and stressful and can result from an event, series of events, or set of circumstances such as a natural disaster, physical or sexual abuse, or chronic adversity (e.g., discrimination, racism, oppression, poverty)" [12]. While the diagnosis of DR-TB and subsequent hospitalization (and thus removal from family and familiar surroundings, often for months at a time) could certainly fit this definition, we utilize instead the term "stressful experiences" throughout this paper. Stress is defined as an experience that is perceived as a threat to wellbeing [13]. The data analysis also utilized a social-ecological model [14] which considers the needs and roles of different actors within the larger societies and communities in which they live.

A thematic network analysis was performed on the study interviews and transcripts [15, 16] with a focus on the experiences of the children and their caregivers. After an initial review of the data during which participants described the experience of their illness and how this affected their life during and after treatment, a coding system was developed by one study team member (JF). This analytic framework was verified/modified by three additional authors (SM, BS, NM), and the interviews analyzed. Discrepancies were resolved via discussion and

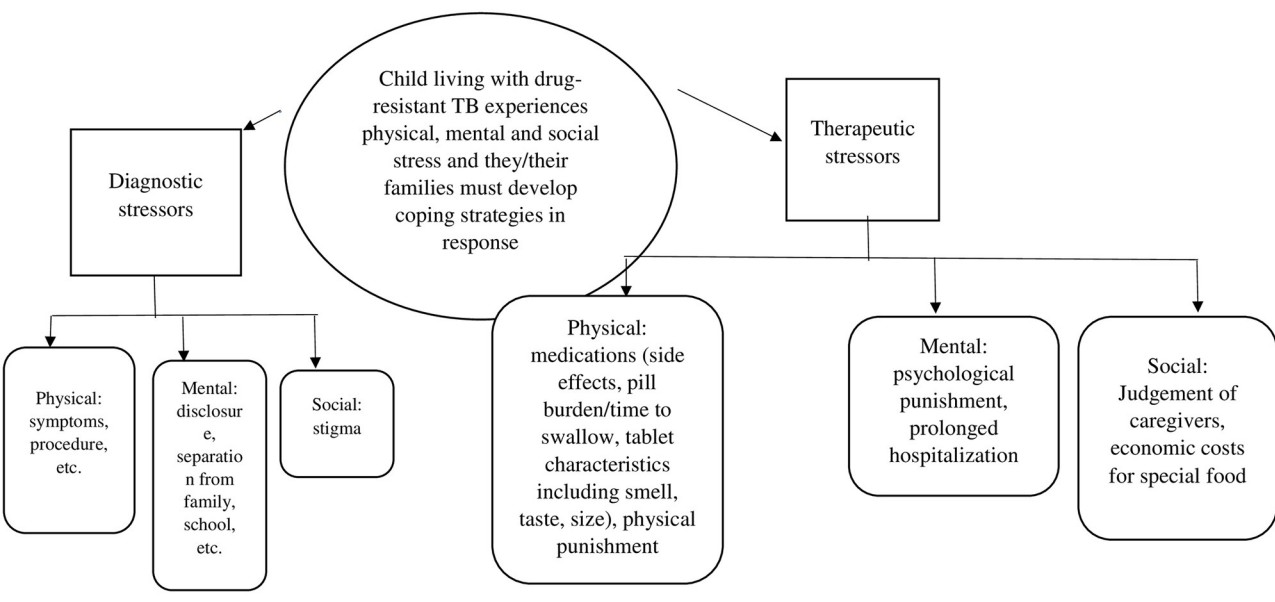

**Fig 1. Analytic framework.**

there was agreement among all study team members on the final analytic framework used, which is presented in Fig 1.

Finally, and as part of the tradition of reflexivity that is essential in doing qualitative research, we note that several of us are engaged in providing care to children with DR-TB as medical providers and this may have impacted our understanding, analysis, and description of the experiences of the children and caregivers who participated in this study. We may have overly focused on biomedical aspects of their experiences. Data collection, analysis, and reporting for this qualitative study followed the consolidated criteria for reporting qualitative research (COREQ) guidelines [17] which can be seen in S3 File.

## Ethics

Written consent was obtained from all the participants over the age of 18 years (for caregivers and children), and all children over the age of 12 years also provided formal oral assent to participate in the study. For children age 12 years and under, only caregiver consent was required, but children only participated if they informally agreed. The consent/assent included participation in the interview and digital audio recording/transcribing/translating, the voluntary terms of involvement in the study and the assurance of confidentiality and anonymity. Patient anonymity was maintained by identifying each patient using a unique identification number. Ethical approval was obtained from the UMgungundlovu Health Ethics Research Board (UHERB). Ref. 002/2020.

## Results

### Demographics

The age, gender, HIV status and education level of the children who participated in this study are included in Table 1.

In terms of caregivers, the demographics are 2 males and 28 females, all over the age of 18 years.

**Table 1. Demographics of the 16 children that participated in this study.**

| Category | No. of Children |
| --- | --- |
| **Gender** | |
| Male | 7 |
| Female | 9 |
| **Age** | |
| 0–6 years old | 5 |
| >6–10 years old | 5 |
| >10–14 years old | 6 |
| **HIV Status** | |
| HIV Positive | 8 |
| HIV Negative | 8 |
| HIV Unknown | 0 |
| **Education Level** | |
| At Home | 4 |
| In School | 12 |
| In Crèche | 0 |
| Unknown | 0 |

In terms of providers, the demographics are 11 females and 1 male, over the age of 18, with experience in pediatric TB ranging between 3 months and 30 years.

## Stressful experiences

In general, it was reported that for a child diagnosed with DR-TB, there is a lived experience of stress that impacts their physical, mental, and social well-being. These pediatric patients and their families therefore develop strategies for coping with these disruptions to their lives. In general, there were major disruptive experiences that resulted from the process around receiving a *diagnosis* of DR-TB and second distinct set of stressful experiences that occurred during the *treatment* of DR-TB once the diagnosis had been made. These two major sub-categories of stress—as well as the coping strategies developed by children and their families to manage them—will be described in more detail below, with a focus on the sub-themes of the physical stress, mental stress, and social stress experienced by participants during the diagnostic and therapeutic periods.

## Diagnostic stress

All of the caregivers and the children talked about a number of difficulties they faced during the period when the child first became sick, and the family was seeking care to determine what was happening to the child. The providers talked less about the stresses faced by children and their families during the diagnostic period, since they usually encountered the children after a diagnosis had already been made.

## Physical

Many of the challenges faced by children and their caregivers involved physical pain either from the DR-TB illness itself or from the diagnostic procedures the child had to undergo to determine what was making the child unwell. As one child participant reported:

"*I was playing at our neighbor's house, and I felt my knees getting weak and painful. Somehow, they managed to take me inside our family home, I couldn't walk, and I felt like I had fever, my temperature was high, and I would also feel pain.*"

And as one caregiver stated:

"*About four in the afternoon she had been playing outside and suddenly fell. She then started to have high temperature and her limbs couldn't move.*"

## Mental

The major mental stresses noted were fear and worry about what was causing the child to experience symptoms, and the mental stresses during the diagnostic period were more likely to be reported by caregivers than by the children themselves. As one caregiver noted:

"*Finding out that my child had the big TB, actually I didn't know that it was TB, he kept losing weight and coughing all the time and I would take him to a clinic in a [nearby] area… They tested him there and the results said it is the big TB and they told me that they would transfer him in this hospital.*"

And another stated:

"*I couldn't bring myself into believing the news I was hearing; I was in denial, and I asked the doctor to please do another test. The doctor refused to do other tests as he confirmed that the results were accurate. Other women who heard me screaming in the doctor's room came to console me and they also couldn't believe that it was true.*"

Although less common, children also reported having anxiety about what was happening to them before they were diagnosed with TB, with one child saying:

"*I was also sad, and I thought I was going to die.*"

Some of the caregivers also reported mental stress and guilt associated with the fact that they themselves were living with DR-TB and felt that they had "passed the disease" on to the child. This belief was often reinforced by the providers and the TB care system. The mental stress experienced by caregivers sometimes changed the ways they interacted with the children, as noted by one caregiver when describing how she talked to her child about DR-TB:

"*I didn't explain anything to her because she was too young to understand, and I had passed it unto her.*"

## Social

The caregivers and children also uniformly reported social challenges that stemmed from their inability to fulfill their usual societal roles, either due to the illness itself or to the time they had to devote to care-seeking. One experience of social stress felt by caregivers during the diagnostic period had to do with costs incurred by the family during the care-seeking process. However, caregivers also reported a notable amount of social stress having to do with stigma associated with TB in their communities. To spare the child from this stigma, sometimes caregivers did not disclose the nature of their child's illness, as one caregiver reported:

"*I don't want to lie I hid it from him that he has TB and because he usually had issues with the health of his eyes, I told him that we are taking him to an eye clinic. I was scared and I did not know how to tell him but at the hospital the nurses really helped me by telling him and when I came to visit him the following week, I discovered that they had already told him.*"

Others reported that the social interactions throughout the diagnostic period were confusing and led to a stressful experience for them and their families, as reported by one caregiver below:

"*My child had flu and I had taken him to a clinic for any typical influenza that children get. The nurse gave us flu medicines and asked me to stop by the TB section and have him checked up as per health precautions medical facilities follow. They did a sputum test and the one circling the wrist area. After a few days, I took him back and the nurse who looked on his wrist said. . .*" take your child home he is a healthy-looking child." *We went back to our normal life, and I sent the child to school. A few days later I get a call and they are asking me "are you his mother" I said yes, I am' . . . within hours they were at my place. They called me to the car and enquired about my child's results and I told them the tale of what had been said to me about the wrist skin reaction being normal and the nursing sister sending us home saying we are clear. They told me that the sputum test result had come back positive for MDR, and I need to stop the child from going to school as they are preparing for his referral to the TB treating hospital. They left and for a week they never called or checked on us again.*"

Although these experiences during the diagnostic period were challenging, a majority of the stressful experiences reported by caregivers and children occurred during the therapeutic period, and these will be discussed in more detail in the next section.

## Therapeutic stress

Once a DR-TB diagnosis was made, rather than entering a period of relief from stress, children and their caregivers reported that they found themselves facing a whole new series of stressors in the physical, mental, and social realms. These challenging experiences were also confirmed by the providers who were caring for the children during this period.

## Physical

The main physical stressors faced by the children had to do with the treatment they received and the tests that had to be done to monitor the child while s/he was receiving treatment. Some of the most difficult physical stress described by the children and caregivers had to do with the multiple, adult tablets the children had to swallow each day. Often it was the inherent characteristics of the medications used to treat the child that led to the challenges, including the number of pills, the size of the pills, and the taste or smell of the pills. As one female caregiver noted:

"*We pray to God that he ends TB and that he does not get it again because all pills are hard to take. I am grateful that now we are going to have another treatment type because the ones he was taking were very huge and hard to swallow and hurting him on the neck which is something he had to take daily.*"

Multiple children commented on the difficulties of having to take adult tablets, especially if those tablets were manipulated in ways that were not intended in order to give them to the

child (i.e., cutting in half of a non-scored tablet to give the child a smaller dose). As one child noted:

> "*To be honest sometimes it was even difficult swallow the one broken into half, it used to get stuck and hurt my throat.*"

And another child reported feeling ill and losing her appetite from the large amounts of liquid she had to drink to "force" the manipulated tablets down her throat, noting:

> "*Because when you are drinking it as pills, they often get stuck here [pointing to throat] and you are then forced to drink lot of water and after drinking so much water you don't feel like taking anything to eat, you just want to sit and let everything subside.*"

The taste and smell of the medications were also a source of stress. As one caregiver reported:

> "*I was using syrups and it was hard because he used to complain and say that they are too bitter, and it is hard to swallow them. . .he used to ask me to give him sweets or something to take them down with.*"

And as one child reported:

> "*The most difficult thing to take was the smell of [one of the medicines] . . .Yes, it smelled so bad.*"

Sometimes it was the adverse events associated with the medications or their administration that led to the physical stress. As one caregiver reported:

> "*He would say it smells bad and when he would take it his stomach would just turn and he would have nausea.*"

Many children confirmed that the adverse events were physically challenging and disruptive to them, as this quote from one child illustrates:

> "*The bad experience for me was that I couldn't eat just after I had taken my medication. . .- When I was back at home and going to school, I was only able to eat during break time. . .. I was hesitant to eat because I didn't want to throw up.*"

So difficult was taking the tablets, in fact, that sometimes, the children reported being physically restrained by people who were administering the medications. As one child reported:

> "*If you didn't want to drink your pills, she would put you on your bed and she would open your mouth and push it in. . .she holds you like this and push it inside you.*"

This was confirmed by the providers, as noted in the exchange below (although they did not provide a direct answer to the question asked by the interviewer):

> "***Interviewer***: *So, there are 2 or 3 of you at a time and is it because you have to hold them*?
>
> ***Provider Participant D***: *Yes, we hold him*

*Interviewer*: *Then when you holding him like that, how long do you hold him*?

*Provider Participant D*: *Yes, some they do take it well but most of the time we have to force it down [other **Provider Participant**s concur in the background]*"

To alleviate some of this physical stress, the children and their caregivers often requested that there be different formulations of the medications that could be given to the children. As one caregiver stated:

"*We would like to request that they replace it with something else because many children suffer with it so much. Preferably a syrup or a tablet that will dissolve in water but definitely not this one because it smells too bad.*"

Families would often try to cope with the bitterness of the medications by masking the taste. As one caregiver noted:

"*My child also took her morning pills with 'Oros' juice. This was helpful, I think because it has that sour taste.*"

Caregivers also reported using practical adherence reminders, such as setting alarms, as noted by one:

"*We would set an alarm as a reminder for all of us that at 03h00 we had to start with this one then wait and start giving the treatment for the day at 05h30AM to prepare for the day as he was heading to school.*"

It was often the child himself/herself who would be person most able to take responsibility for taking the medication. As one caregiver reported:

"*She used to be able to count her pills and she knew their amount. She would even remember the one she needed to take on a Wednesday. I would watch her with awe as she swallowed the first handful, take a breath and then follows with the rest. My job was to ask her if she was doing okay, and she would say that yes, she is.*"

## Mental

The stressful mental experiences faced by the children and their caregivers stemmed from two major sources. The first were the techniques used by providers and by caregivers to ensure that children took their DR-TB treatment. As noted earlier, the treatment of DR-TB involves a combination of medications—as many as seven different drugs in some instances—and children usually have to take more than one tablet of each kind of medication, leading to a high pill burden. This pill burden was a source of anxiety and mental suffering for the children undergoing treatment, as one child noted:

"*There were a lot of medicines to take and sometimes I couldn't take and drink them, sometimes the treatment would make my heart to beat fast because there was just a lot to take.*"

In order to help cope with the large pill burden that characterized treatment, caregivers and children developed multiple strategies to help lessen the stress. For example, as one caregiver noted:

"*He used to ask me not to give them one by one, so I use to give him maybe three at a time.*"

Because most children are treated with adult formulations of these medications, daily consumption of treatment is a grueling challenge, and it has done for a period of nine to 18 months. Thus, it is not surprising that caregivers and providers resort to a variety of techniques to ensure the children consume their medication each day, including the threat of punishment. While understandable, however, such practices around medication administration can lead to mental stress for children. Children reported that "scare" tactics were often used to get them to take their tablets, as exemplified by the quote below:

""*If you cry and say you want to be with your mum [she] would [say she was going to] take you up to the top to sit with the snakes.*"

Caregivers confirmed the use of such coercive measures, with one noting:

"*Yes, you would have to scare the child and say a bug will catch you if you don't take the treatment and the doctor will come here to inject you. . .there's a certain monkey I don't know what they used to call it, I'd just say 'here's that monkey' and he would take the medication without hesitation. I'd scare him and say, 'I'm going to take you out through the window and take you to that certain monkey'.*"

In contrast to this and as a comparative coping strategy, some caregivers reported using encouraging words or playful concepts with the children. As one caregiver reported:

"*I used to tell him that 'bro, if you want to get better there is nothing else, we can do and the best way is that you need to take your meds. . .it is time . . .this is about your life boy-boy, come it is time for your meds'. . .yes.*"

And as another noted:

"*I would even ask to check his pockets and say I want to see if one didn't fall in your pockets by mistake and then I would commend him and tell him he has done well by drinking his pills.*"

Caregivers also reported "rewarding" children at times, as noted below:

"*He knew what to do and I would also tell him that this about his own life and I encouraged him that if he took his medication well, he will be cured, and he accepts and understands that. Also, at times when I have some money, I do buy him some goodies.*"

Some caregivers did report, however, that the children were able to take treatment without much input from them, as noted below:

"*I just call him by his name and say. . .'come it's time to take your medication' and he just come to me without any hassle. Then I put them on a saucer and place each pill and as I finish placing them, he takes the entire lot, put them in his mouth and swallow them with water and then he is okay.*"

The second source of mental stress for children and their caregivers was the prolonged period of hospitalization that was standard for children undergoing treatment for DR-TB. All

of the children in this study were hospitalized in the centralized DR-TB hospital in order to initiate treatment. This period of hospitalization—during which the child stayed in the hospital without his or her family—lasted anywhere from several weeks to several months and was a major source of trauma for the children and their families. As one child noted:

"*There was that gap in my heart of missing my family back at home.*"

And as one caregiver reported:

"*In my mind I thought I was going to stay in hospital with him and so I packed clothes in a suitcase for both of us. We were taken by an ambulance. . ..and to my surprise that when we got here, they told me that I am leaving my child behind. I think that was the hardest moment ever.*"

So difficult was the hospitalization, in fact, that when a child was finally discharged home, parents encouraged adherence to therapy by telling the child he or she would have to go back to the hospital if he or she did not take treatment, as shown in the quote from one caregiver below:

"*We would try and plead with her at home. We have other small children the same age as her and when she returned from the hospital, we were preparing for the other children to start school–grade R. We would tell her that if you comply, you see we will also buy you your school bag and you can go to school like your siblings, and we would tell her that she will not be going back to the hospital if she drinks her treatment well.*"

The reasons for hospitalization often had to do with the complexity of the treatment regimen, although some children were also too clinically ill to remain at home. While the large number of tablets that had to be given each day were one reason providers reported, they felt more comfortable treating the children in the inpatient setting. The lack of child-friendly formulations of the second-line drugs were also reported to be a contributing factor to prolonged hospitalization in the study population. As one health care provider reported:

"*Even if you discharge for the parent, they don't have this thing that they have to grind them and grinding them is very inconveniencing and makes fingers sore. They also don't have the things you use for grinding.*"

Some providers reported that they felt they could ensure adherence better if the child remained in the hospital, and they sometimes felt they did a better job taking care of the children than the families and caregivers. As one health care provider reported:

"*I enjoy working with babies, the only thing I hate is their 'defaulting mothers', you find that children receive treatment, but their parents don't consistently give them in way that they've been prescribed which causes a child to be sick then they have to come back for the same issue without any progress.*"

It was clearly frustrating for the providers to care for children who had not completed their treatment or had to be re-admitted. They also seemed to blame the caregivers for non-adherence to treatment.

However, if the providers were able to be thoughtful and open in their discussions with family members, this helped the caregivers cope with the period of hospitalization. As one caregiver reported:

"*I remember how sick and how thin he was before coming to the hospital, his skin was peeling but the joy he had after a few weeks in hospital is indescribable. He had gained weight and was coping very well with pills. The hospital staff also treated us with respect and dignity and allowed us visiting time without restrictions.*"

Some caregivers also reported that even though the prolonged hospitalization was difficult, they felt it was important for the child's health in the long term. As one caregiver stated:

"*I can say things were easier with my nephew. He spent a lot of time in hospital, and they had coached him well. Coming to visit him early after his admission there were some challenges but by the time, he was being discharged he was the master of adherence. It was amazing to see that he even knew their names, imagine as adults we even forget the names of these pills but here's a young man who could tell me what each is called and so I knew that should it happen that I make a mistake he would correct me, and it encouraged me to see him young as he is but taking his treatment seriously.*"

Others reported that they learned how to care for the child by seeing what was done in the hospital and then replicating it at home. As one caregiver reported:

"*With my child. . . I decided to follow the routine she had learned here in hospital. In August I came here to see a social worker and I was able to peep at the ward and saw how they were administering medications. Upon her being discharged I was adamant that I was going to follow the 8am routine. Here, she would first take treatment and wait for about 30 minutes to allow medication to settle and after that time she would have something to eat.*"

Another method used by families for dealing with the mental stress was religion. Although this was not reported by a majority of caregivers in the study, those that did report using religion to help cope with the mental stresses of a child undergoing treatment for DR-TB felt it was a tremendous source of relief, as noted in the quote from a caregiver (who was being treated for DR-TB in the same hospital as her child) below:

"*My anxiety heightened whenever I would see a hearse. People were dying every day and we lived in so much fear. I could see a hearse from [my hospital bed] and it would pass by the children's ward where my child was staying, and I would be gripped with fear thinking it was my child. Prayer and reading the Bible was a pillar and kept me going.*

## Social

The continued exclusion from usual social roles was an ongoing source of stress for children with DR-TB and their caregivers. So great was the disruption in his normal functional status, that one child reported:

"*Pills were my whole life.*"

Families also reported that the stigma of TB in the community was also a source of social stress for them once the child was diagnosed with DR-TB. AS one caregiver reported:

"*The most difficult part of our journey was when he started taking treatment. It came with its own challenges. None of my family members had had TB before. I had to go back and tell*

*them that my child has MDR TB, and we all had to get tested as a family. We are a big family it is was not feasible for everyone to get tested and eventually some went one by one but not everyone in our family got tested.*"

Another reported that their child faced notable discrimination because of having a DR-TB diagnosis:

"*Mine was stigmatized and. . .they used to say, 'he is taking pills' and get questioned what those pills are for and reply saying, 'they are for TB'. He was even stopped from attending [school] as they said he was going to infect other children. . .*"

Because of stigma and discrimination, caregivers and children sometimes felt isolated and alone in their treatment journey, as one caregiver reported:

"*I am saying that no one was there for me, it was just myself and my mother who took the responsibility for my child and even with my extended family no one cared or bothered to even visit my child whilst they were in hospital.*"

So great was the social isolation children faced, that some of them reported that they preferred to remain in the hospital, relying on the bonds they forged with other children there to help them cope. One child reported:

"*It was better [at the hospital than home] because you didn't feel like you are the only one that is taking medication, and I would just tell myself that all of us are in this whereas at home it is just only me.*"

This group support was confirmed by a provider, who noted there was also a competitive element to adherence when there was more than one child taking treatment:

"*To others it's like competition, if you see the other one takes it, you will also take it.*"

The main coping strategy described by families for dealing with stigma had to do with showing the child extra love and support. As one caregiver noted:

"*TB—that does not mean that your child cannot play with other children. Love your child and make him still feel as a member of your family. Know that he is not different and don't discriminate against him, your child should feel love.*"

Caregivers also reported that the providers at the hospital were an important source of support for them. As one caregiver noted:

"*I really want to extend my gratitude to the TB hospital unit; they really took good care of my child. . . His father and I would come and visit him, and he was making good progress.*"

Other families reported that they had to give up some of their usual activities in order to prepare the medications for the child to take once the child was released from the hospital and returned home. So much work was involved in the medication preparation, in fact, that even when the child was well enough to return to school, the families could not re-engage in normal activities. As one caregiver reported:

"*With my grandchild I use to wake up at 4am and crush pills for him. He could not drink a solid pill, so I had to crush them and put them in water then wake him up at 4.30am for him to drink. It was hard; I had to apply patience until he was finished.*"

In terms of other social stressors, families reported that they continued to face financial burdens during the treatment period. Some of this came from the costs associated with visiting the child in the hospital. As one caregiver reported:

"*I was forced to leave my work, and another difficult thing I was faced with was that I was three months pregnant at the time. I used to sleep on hospital benches. . . I had to walk a long distance before I could get public transport so I could come and visit him.*"

And another reported:

"*In the early days of his hospital admission he used to cry a lot and he couldn't be distanced from me. I was fortunate that at that time I was receiving a social grant for myself. . . I used to take the very same money which I was meant to contribute towards my family's food groceries and use it for the transport fee all the way from [our village] to come and see my child.*"

Other financial difficulties had to do with the need for special food purchases, many of which went beyond what could be accommodated by the family budget. Sometimes, the families sought help from providers to address these financial issues, but they reported that such requests for help were often not met, leaving the family in difficult financial straits. An example provided by one caregiver illustrates this:

"*I tried to ask the doctor to endorse for me to get a government child support grant, but he refused and said I should manage with what I've got. I tried to tell him that my child liked meat because what I noticed is that these pills increase children's appetite for food, and he said liking meat is not enough reason for endorsing a grant. I raised the point of needing to buy such things as the yogurt and juice, but he adamantly denied and said I should just come up with a plan to make my child drink pills and use the child support grant I am receiving already. He said I am the child's mother, and I should find a way to make it with the little that I am getting and to know that this is for my child's health, and it doesn't matter what I do but I should find a way.*"

## Discussion

This qualitative study of the diagnostic and treatment experiences of children and caregivers of children undergoing pediatric DR-TB treatment revealed multiple sources of physical, mental, and social stress encountered during the diagnostic and therapeutic periods. While some of these stressors are inherent in the illness itself and similar to what has been reported in the diagnosis and treatment of children with other serious illness [18, 19] others are unique to DR-TB. And while many are also similar to what is experienced by adults with DR-TB [20], children with this disease also face unique challenges. In the diagnostic arena, for example, the reliance on insensitive, sputum-based technologies that cannot be implemented at a primary care level means that families face notable stress during the diagnosis, including financial costs [21] as was reported in this study.

In terms of the therapeutic stressors reported by children, caregivers, and providers, many could be addressed by providing community-based, child-friendly services. Participants

**Table 2. Summary of pediatric DR-TB stressful experiences and possible approaches to address them.**

| Category of stressor | Specific Stressful Experiences | Possible Mitigation Strategies |
|---|---|---|
| *Diagnostic Physical* | Physical symptoms due to illness | Earlier diagnosis of TB |
| | Pain from diagnostic tests and procedures | Perform diagnostic testing on samples that are easy to obtain |
| *Diagnostic Mental* | Anxiety/fear and depression about illness | Counseling and support |
| | Self-blame around transmission | Counseling and support |
| *Diagnostic Social* | Stigma/discrimination | Community education, advocacy, counseling and support |
| | Financial burdens associated with treatment | Incentives and enablers for TB diagnosis |
| *Therapeutic Physical* | Consumption of large number of pills, especially if they are cut or manipulated versions of adult tablets | Child-friendly formulations of medications, combination tablets, shorter regimens with fewer drugs |
| | Taste/smell of the tablet | Taste-masked, smell-masked medications |
| | Adverse events associated with the tablet | Safer and more tolerable medications |
| | Monitoring tests | Safer medications, perform tests on samples that are easy to obtain |
| | Physical punishment/restraint associated with medication administration | Supportive/encouraging measures of adherence, adherence incentives, child-friendly formulations of medications, combination tablets, shorter regimens with fewer drugs |
| *Therapeutic Mental* | Fear/coercion used to ensure children take the medication | Supportive/encouraging measures of adherence, adherence incentives, child-friendly formulations of medications, combination tablets, shorter regimens with fewer drugs |
| | Anticipatory anxiety/fear about possible side effects | Counseling and support |
| | Isolation due to prolonged hospitalization/separation from families/home | Community-based & decentralized treatment, peer support groups, counseling and support, school programs in hospitals for children who cannot be treated in the community |
| *Therapeutic Social* | Exclusion from normal roles | Evidence-based infection control practices, community-based & decentralized treatment |
| | Stigma/discrimination | Community education, advocacy, counseling and support, peer support groups |
| | Financial burdens associated with treatment | Incentives and enablers for TB treatment |

described therapeutic stressors that resulted from the outdated models of treatment that unfortunately seem to characterize pediatric DR-TB practices, including prolonged hospitalization (which may disrupt important developmental activities, such as schooling [22]), reliance on adult formulations of the second-line drugs, coercive adherence strategies, and prolonged social exclusion for children diagnosed with the disease [23]. Table 2 summarizes these sources of stress as reported by participants as well as potential approaches that could be used to address them. It is noteworthy that having more child-friendly formulations of the medications used to treat pediatric DR-TB could alleviate many of the challenges reported by the children and their caregivers in this study. Such formulations have finally been developed and are available for use [24], but their limited uptake may jeopardize their accessibility in the future [25].

In spite of these numerous stressful experiences, children and their caregivers showed a remarkable amount of resilience as they navigated their way through the pediatric DR-*TB* diagnostic and treatment landscape. Many of them had developed compensatory coping mechanisms that allowed them to maintain some semblance of normalcy during the diagnostic and treatment periods, and they were often supported by a team of providers during their difficult journeys. These effective strategies could also be incorporated into more child-friendly models of service delivery when it comes to DR-TB and form a "grassroots" compendium of "best practices" that could be supported systematically.

There are multiple limitations to this study. First was the notable number of children treated at the hospital where neither the child nor the caregiver participated in the study. There were multiple reasons for this, but it may have led to a biased sample. The research team made every effort to support participation, but usual recruitment efforts were hampered by the COVID-19 crisis. Second, the study relied on focus groups as the primary means for collecting

data from participants. Focus groups are a standard method for obtaining qualitative data and they can be an efficient way to collect data from multiple participants. The timelines of this study were shortened due to the COVID-19 pandemic (during which qualitative data collection was put on hold in South Africa) and this is why focus groups were chosen as the data collection method of choice. Using a group, however, could have led to participants reporting experiences that they themselves did not have because of group dynamics. Participants may have been less likely to report "outlier" behaviors if they felt these were not socially acceptable, and thus there may have been important experiences and coping strategies that were not uncovered by this study. While attempts to minimize power differentials by having separate focus groups for providers, caregivers, and children, participants still may have felt intimidated to report on their actual experiences and practices in a group setting. Very young children (i.e., those below the age of 7 years) were also not able to be included in the FGDs. Our practice of reflexivity leads us to acknowledge another limitation of this study in that our work as TB providers (some authors are practicing clinicians as well as researchers) may have led to a focus on biomedical aspects of stressful experiences, both in the analysis of the data and in our conclusions/recommendations. Another limitation is that the numbers of participants were relatively small, and all were receiving care at one referral hospital in South Africa. There was also a notable absence of male caregiver participation, although this likely reflects the actual caregiver demographics in this population. For these reasons, the findings may not be generalizable to other populations of children and their caregivers. Additional studies are needed to confirm these findings, but it is hoped this work can inform such future endeavors.

## Conclusion

In spite of these limitations, this study done among children being treated for DR-TB and their caregivers (with supplemental information given by their healthcare providers) shows that their diagnostic and therapeutic experiences are characterized by physical, mental, and social stress. Addressing these stressors through child-friendly DR-TB testing, treatment, and support is not only essential for ending TB but also for enacting a human-rights based approach to child health in general. Children with DR-TB are a vulnerable population, and they have often been the last to benefit from advances in general pediatric care and in DR-TB care more specifically. Our data show the potentially damaging consequences of this and call for brave actions to be taken by the TB community to urgently address this untenable situation. Such actions should be modeled on what has been accomplished by these "awe-inspiring" children and their families in facing the scourge of DR-TB.

## Supporting information

**S1 File. Interview guide.**
(DOCX)

**S2 File. Inclusivity in global research statement.**
(DOCX)

**S3 File. COREQ checklist.**
(PDF)

## Acknowledgments

We are thankful to the children, caregivers and providers who generously gave of their time to participate in this study. We are hopeful their experiences can help improve care for other

children and families in the future. We are thankful to colleagues in the Stop TB Partnership's Global Drug Facility both for their comments on this manuscript but even more so for the heroic work they do to improve the care of children with drug-resistant TB, with notable work being done by Brian Kaiser, Brenda Waning, and Ramon Crespo.

## Author Contributions

**Conceptualization:** Shriya Misra, Nirupa Misra, Boitumelo Seepamore, Kerry Holloway, Nalini Singh, Jacqui Ngozo, Vusi Dlamini, Jennifer Furin.

**Data curation:** Shriya Misra, Nirupa Misra, Boitumelo Seepamore, Kerry Holloway, Jennifer Furin.

**Formal analysis:** Shriya Misra, Nirupa Misra, Boitumelo Seepamore, Jennifer Furin.

**Funding acquisition:** Shriya Misra, Nirupa Misra, Jennifer Furin.

**Investigation:** Shriya Misra, Nirupa Misra, Boitumelo Seepamore, Jennifer Furin.

**Methodology:** Nirupa Misra, Boitumelo Seepamore, Jennifer Furin.

**Project administration:** Nirupa Misra.

**Validation:** Jennifer Furin.

**Writing – original draft:** Shriya Misra, Nirupa Misra, Boitumelo Seepamore, Jennifer Furin.

**Writing – review & editing:** Shriya Misra, Nirupa Misra, Boitumelo Seepamore, Kerry Holloway, Nalini Singh, Jacqui Ngozo, Vusi Dlamini, Zanele Radebe, Norbert O. Ndjeka, Jennifer Furin.

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
