## [Decision Letter · Decision Letter 0]

18 Jul 2022

PONE-D-22-03236Title : “I would watch her with awe as she swallowed the first handful”: a qualitative study of multidrug-resistant tuberculosis experiences among children, caregivers, and pediatric health providers in Durban, South AfricaPLOS ONE

Dear Dr. Furin,

Thank you for submitting your manuscript to PLOS ONE. After careful consideration, we feel that it has merit but does not fully meet PLOS ONE’s publication criteria as it currently stands. Therefore, we invite you to submit a revised version of the manuscript that addresses the points raised during the review process.

Specifically, the reviewers have multiple concerns including the missing methodology details and the English language usage. Please have all the comments addressed point-by-point.

We look forward to receiving your revised manuscript.

Kind regards,

Jianhong Zhou

Staff Editor

PLOS ONE

3. During your revisions, please note that a simple title correction is required: change current title "Title : “I would watch her with awe as she swallowed the first handful”: a qualitative study of multidrug-resistant tuberculosis experiences among children, caregivers, and pediatric health providers in Durban, South Africa". to "“I would watch her with awe as she swallowed the first handful”: a qualitative study of multidrug-resistant tuberculosis experiences among children, caregivers, and pediatric health providers in Durban, South Africa" Please ensure this is updated in the manuscript file and the online submission information.

4. Please include your tables as part of your main manuscript and remove the individual files. Please note that supplementary tables (should remain/ be uploaded) as separate "supporting information" files.

“This work was supported by a grant from the Stop TB Partnership's Global Drug Facility.”

“This work was supported by a grant from the Stop TB Partnership.  We are thankful to the children, caregivers and providers who generously gave of their time to participate in this study.  We are hopeful their experiences can help improve care for other children and families in the future.  We are thankful to colleagues in the Stop TB Partnership’s Global Drug Facility both for their comments on this manuscript but even more so for the heroic work they do to improve the care of children with drug-resistant TB, with notable work being done by Brian Kaiser, Brenda Waning, and Ramon Crespo.”

“This work was supported by a grant from the Stop TB Partnership's Global Drug Facility.”

7. Thank you for stating the following in your Competing Interests section: 

“The authors have no competing interests to declare.”

8. In your Data Availability statement, you have not specified where the minimal data set underlying the results described in your manuscript can be found. PLOS defines a study's minimal data set as the underlying data used to reach the conclusions drawn in the manuscript and any additional data required to replicate the reported study findings in their entirety. All PLOS journals require that the minimal data set be made fully available. For more information about our data policy, please see http://journals.plos.org/plosone/s/data-availability.

9. We note that you have indicated that data from this study are available upon request. PLOS only allows data to be available upon request if there are legal or ethical restrictions on sharing data publicly. For more information on unacceptable data access restrictions, please see http://journals.plos.org/plosone/s/data-availability#loc-unacceptable-data-access-restrictions.

10. Please include captions for your Supporting Information files at the end of your manuscript, and update any in-text citations to match accordingly. Please see our Supporting Information guidelines for more information: http://journals.plos.org/plosone/s/supporting-information.

Reviewers' comments:

Reviewer's Responses to Questions

**Comments to the Author**

1. Is the manuscript technically sound, and do the data support the conclusions?

Reviewer #1: Partly

Reviewer #2: No

Reviewer #3: Yes

2. Has the statistical analysis been performed appropriately and rigorously? 

Reviewer #1: N/A

Reviewer #2: N/A

Reviewer #3: N/A

3. Have the authors made all data underlying the findings in their manuscript fully available?

Reviewer #1: Yes

Reviewer #2: No

Reviewer #3: Yes

4. Is the manuscript presented in an intelligible fashion and written in standard English?

Reviewer #1: No

Reviewer #2: No

Reviewer #3: Yes

5. Review Comments to the Author

Reviewer #1: This is an interesting argument and delves into a neglected area of research on children and tuberculosis from a qualitative perspective. The manuscript describes it methods well and their use of grounded theory to examine the data. Their data is strong. The article needs a good copy edit as their are mistakes contained in it.

Firstly, I struggled with the argument that what children struggle with is trauma. Trauma is defined as a specific psychological condition and the evidence does not point to these children or their parents experiencing trauma. I quote from the manuscript: "In general, it was reported that for a child diagnosed with DR-TB, there is a “lived experience of trauma that impacts their physical, mental, and social well-being”. " However trauma is defined as "an emotional response to a terrible event like an accident, rape or natural disaster. Immediately after the event, shock and denial are typical. Longer term reactions include unpredictable emotions, flashbacks, strained relationships and even physical symptoms like headaches or nausea. While these feelings are normal, some people have difficulty moving on with their lives." The authors need to find more accurate terms or phrases to describe what the evidence is telling them - disturbances, dissonance, inconveniences, disruptions, changes in life worlds etc.

Secondly, the main thing that struck me was that what children experience is almost identical to what adults experience barring several issues. This appears very relevant to me as a finding. Diagnosis is difficult, initial misdiagnosis, stigma, shock, taking the medication, the smell of certain medications, drinking of Oros, the financial struggles, hospitalisation and the ways that individuals are expected to become responsible for their well being etc. These are well documented in social science research. What stands out as exceptions are their schooling, but there is some evidence that hospital schools benefit children from under privileged backgrounds - references below, and the reliance on adults (lack of autonomy) to determine their health care or lack of it. This feels like a finding that could be brought into the article and would make the most relevant argument than that of pursuing ideas of trauma that cannot be proved.

I include two references from UCT Anthropology that have worked specifically with children and tuberculosis. I can contacted for a copy of these. helen.macdonald@uct.ac.za

Schooling, Tarryn 2014. “Learning Interrupted: How a TB Diagnosis Affects Education” Unpublished Honours, Social Anthropology, UCT.

Abney, Kate 2014. At the Foot of Table Mountain: Paediatric Tuberculosis Patient Experiences in a Centralised Treatment Facility in Cape Town, South Africa, Unpublished PhD, Anthropology, UCT

Reviewer #2: The title is very attractive . The analysis and interpretation do not go in sync.

It is unclear how the respondents categorized as children be 0-14yrs. How did the team even think of eliciting information through FGDs from this group . There is no mention of how many within each group and it is unclear how babies- or even those older could participate in FGDs.

The author highlights the caregivers and hardly any information from providers.

It would be better for the authors to confine to care givers and their experiences in dealing with the pediatric group and their challenges befitting the title

The other groups need not be covered .

The recommendations could be for the Health care providers and policy makers.

Very vague analysis and while the recommendations and suggestions are good this does not come from the analysis of this nature

Please revise and resubmit

Reviewer #3: The authors identify a significant gap in the literature. This is an important area to study because of the potential for direct impact on care of children with DR-TB in certain SA care contexts and more broadly in terms of awareness-making around a lack of paediatric treatment formulations. This is a strong piece of research, but my main concern is a lack of theoretical basis for conceptualising trauma. The authors take it for granted that the reader knows which conception(s) of trauma are being used for analysis of the primary interview data. It is essential that it be made clear (1) what is meant by trauma; (2) clarify the different figurations of trauma with reference to theory as these figurations structure the findings; (3) place the study and its findings within a clearer theoretical literature - if trauma is the focus, the background literature should reflect this.

METHODS

- It would be good to have more detail on what constituted a 'caregiver' and a 'provider' in the context of the study and how this informed selection.

- Line 141: typo/syntactical issue

- Were transcripts transcribed by researchers or externally? Important to know because of implications for data handling and ethics.

- How was the study informed by Das and Colleagues framework? Brief explanation and justification (1 sentence).

- Reflexivity: how would the authors' positions as clinicians affect interpretation? How were reflexivity issues incorporated into findings/discussion (why relevant)?

BACKGROUND

- Excellent, justified the need for this research.

ETHICS

- How was consent gained for participants younger than 12? Did their guardians consent?

FINDINGS

- Needs a working definition/theoretical basis for what constitutes trauma.

- 320-322: Provider Participant D does not respond directly to question. Does this reflect the actual transcript? If so, it might be worth mentioning that the response wasn’t direct.

6. PLOS authors have the option to publish the peer review history of their article (what does this mean?). If published, this will include your full peer review and any attached files.

Reviewer #1: No

Reviewer #2: No

Reviewer #3: No

---

## [Author Response · Author response to Decision Letter 0]

20 Jul 2022

We have confirmed and edited this accordingly.

We have completed this questionnaire and included it with our submission files.

3. During your revisions, please note that a simple title correction is required: change current title "Title : “I would watch her with awe as she swallowed the first handful”: a qualitative study of multidrug-resistant tuberculosis experiences among children, caregivers, and pediatric health providers in Durban, South Africa". to "“I would watch her with awe as she swallowed the first handful”: a qualitative study of multidrug-resistant tuberculosis experiences among children, caregivers, and pediatric health providers in Durban, South Africa" Please ensure this is updated in the manuscript file and the online submission information.

We have changed the title of the paper to reflect the comments of the reviewers and have added the new title to the documents. 

4. Please include your tables as part of your main manuscript and remove the individual files. Please note that supplementary tables (should remain/ be uploaded) as separate "supporting information" files.

Thank you. We have included the tables in the text.

“This work was supported by a grant from the Stop TB Partnership's Global Drug Facility.”

We have done this.

“This work was supported by a grant from the Stop TB Partnership. We are thankful to the children, caregivers and providers who generously gave of their time to participate in this study. We are hopeful their experiences can help improve care for other children and families in the future. We are thankful to colleagues in the Stop TB Partnership’s Global Drug Facility both for their comments on this manuscript but even more so for the heroic work they do to improve the care of children with drug-resistant TB, with notable work being done by Brian Kaiser, Brenda Waning, and Ramon Crespo.”

“This work was supported by a grant from the Stop TB Partnership's Global Drug Facility.”

We have removed the funding statement from the paper and have added this information to the cover letter and to the funding statement. 

7. Thank you for stating the following in your Competing Interests section: 

“The authors have no competing interests to declare.”

We have included this information in the cover letter.

8. In your Data Availability statement, you have not specified where the minimal data set underlying the results described in your manuscript can be found. PLOS defines a study's minimal data set as the underlying data used to reach the conclusions drawn in the manuscript and any additional data required to replicate the reported study findings in their entirety. All PLOS journals require that the minimal data set be made fully available. For more information about our data policy, please see http://journals.plos.org/plosone/s/data-availability.

Please note that we did provide a statement in the supplemental files on the fact that there were ethical restrictions put on our data such that it could not be shared. We have added this information to the cover letter, and it is copied in below.

“Data collected in this study were from a small number of participants and the focus group discussions contain potentially identifiable data. Thus, the ethics review board did not approve making the data widely available on open access. Thus, there are ethical restrictions to making this data available since it contains identifying information. We have provided our interview guides and coding framework. If there are any requests to view the data, they can be sent to the UMgungundlovu Health Ethics Research Board in Durban, South Africa by contacting the chairperson and administrator of the ERB at: Chairperson: Dr Damian Clarke Email: damianclar@gmail.com; Administrator: Nqobile Makhathini (033 395 2102) Email: nqobile.makhathini@kznhealth.gov.za”.

9. We note that you have indicated that data from this study are available upon request. PLOS only allows data to be available upon request if there are legal or ethical restrictions on sharing data publicly. For more information on unacceptable data access restrictions, please see http://journals.plos.org/plosone/s/data-availability#loc-unacceptable-data-access-restrictions.

There were ethical restrictions placed on the data because it could be identifiable. Data collected in this study were from a small number of participants and the focus group discussions contain potentially identifiable data. Thus, the ethics review board did not approve making the data widely available on open access. Thus, there are ethical restrictions to making this data available since it contains identifying information. We have provided our interview guides and coding framework. If there are any requests to view the data, they can be sent to the UMgungundlovu Health Ethics Research Board in Durban, South Africa by contacting the chairperson and administrator of the ERB at: Chairperson: Dr Damian Clarke Email: damianclar@gmail.com; Administrator: Nqobile Makhathini (033 395 2102) Email: nqobile.makhathini@kznhealth.gov.za

There were ethical restrictions placed on the data because it could be identifiable. Data collected in this study were from a small number of participants and the focus group discussions contain potentially identifiable data. Thus, the ethics review board did not approve making the data widely available on open access. Thus, there are ethical restrictions to making this data available since it contains identifying information. We have provided our interview guides and coding framework. If there are any requests to view the data, they can be sent to the UMgungundlovu Health Ethics Research Board in Durban, South Africa by contacting the chairperson and administrator of the ERB at: Chairperson: Dr Damian Clarke Email: damianclar@gmail.com; Administrator: Nqobile Makhathini (033 395 2102) Email: nqobile.makhathini@kznhealth.gov.za

10. Please include captions for your Supporting Information files at the end of your manuscript, and update any in-text citations to match accordingly. Please see our Supporting Information guidelines for more information: http://journals.plos.org/plosone/s/supporting-information.

We have added these to the text.

Reviewers' comments:

Reviewer's Responses to Questions

Comments to the Author

1. Is the manuscript technically sound, and do the data support the conclusions?

Reviewer #1: Partly

Reviewer #2: No

Reviewer #3: Yes

2. Has the statistical analysis been performed appropriately and rigorously?

Reviewer #1: N/A

Reviewer #2: N/A

Reviewer #3: N/A

3. Have the authors made all data underlying the findings in their manuscript fully available?

The PLOS Data policy requires authors to make all data underlying the findings described in their manuscript fully available without restriction, with rare exception (please refer to the Data Availability Statement in the manuscript PDF file). The data should be provided as part of the manuscript or its supporting information or deposited to a public repository. For example, in addition to summary statistics, the data points behind means, medians and variance measures should be available. If there are restrictions on publicly sharing data—e.g. participant privacy or use of data from a third party—those must be specified.

Reviewer #1: Yes

Reviewer #2: No

Reviewer #3: Yes

4. Is the manuscript presented in an intelligible fashion and written in standard English?

Reviewer #1: No

Reviewer #2: No

Reviewer #3: Yes

5. Review Comments to the Author

Reviewer #1: This is an interesting argument and delves into a neglected area of research on children and tuberculosis from a qualitative perspective. The manuscript describes it methods well and their use of grounded theory to examine the data. Their data is strong. 

Thank you for this review and these helpful comments.

The article needs a good copy edit as their are mistakes contained in it.

We have done a thorough copy edit of the paper and corrected mistakes. We apologize for the errors. 

Firstly, I struggled with the argument that what children struggle with is trauma. Trauma is defined as a specific psychological condition and the evidence does not point to these children or their parents experiencing trauma. I quote from the manuscript: "In general, it was reported that for a child diagnosed with DR-TB, there is a “lived experience of trauma that impacts their physical, mental, and social well-being”. " However trauma is defined as "an emotional response to a terrible event like an accident, rape or natural disaster. Immediately after the event, shock and denial are typical. Longer term reactions include unpredictable emotions, flashbacks, strained relationships and even physical symptoms like headaches or nausea. While these feelings are normal, some people have difficulty moving on with their lives." The authors need to find more accurate terms or phrases to describe what the evidence is telling them - disturbances, dissonance, inconveniences, disruptions, changes in life worlds etc.

Thank you for this comment. We had been relying on the analytic framework by Das and colleagues for adolescent DR-TB, which specifically described these experiences as trauma (see Das M, Mathur T, Ravi S, et al. Challenging drug resistant TB treatment journey for children, adolescents and their caregivers: A qualitative study. PLoS ONE 2021, 16(3): e0248408). We had also been referring to the literature showing a high degree of “trauma” among children hospitalized with other life-threatening diseases, such as cancer. In this literature, they describe trauma as physical and psychological experiences that are distressing, emotionally painful, and stressful and can result from an event, series of events, or set of circumstances such as a natural disaster, physical or sexual abuse, or chronic adversity (e.g., discrimination, racism, oppression, poverty).” While we did feel the data supported the use of this term “trauma” we apologize for not explaining it well enough. We did take this reviewer’s comment seriously and have changed the term “trauma” to “stressful”, “stressors” and/or “stressful experiences” throughout the paper. We did, however, leave the term “trauma” in the methods since the framework we used to develop the analysis specifically used the term trauma. 

Secondly, the main thing that struck me was that what children experience is almost identical to what adults experience barring several issues. This appears very relevant to me as a finding. Diagnosis is difficult, initial misdiagnosis, stigma, shock, taking the medication, the smell of certain medications, drinking of Oros, the financial struggles, hospitalisation and the ways that individuals are expected to become responsible for their wellbeing etc. These are well documented in social science research. What stands out as exceptions are their schooling, but there is some evidence that hospital schools benefit children from under privileged backgrounds - references below, and the reliance on adults (lack of autonomy) to determine their health care or lack of it. This feels like a finding that could be brought into the article and would make the most relevant argument than that of pursuing ideas of trauma that cannot be proved.

I include two references from UCT Anthropology that have worked specifically with children and tuberculosis. I can be contacted for a copy of these. helen.macdonald@uct.ac.za

Schooling, Tarryn 2014. “Learning Interrupted: How a TB Diagnosis Affects Education” Unpublished Honours, Social Anthropology, UCT.

Abney, Kate 2014. At the Foot of Table Mountain: Paediatric Tuberculosis Patient Experiences in a Centralised Treatment Facility in Cape Town, South Africa, Unpublished PhD, Anthropology, UCT

Thank you very much for this comment. We added a sentence in the discussion noting the similarity with adult DR-TB patient experiences. The discussion now reads (lines 579-580):

“And while many are also similar to what is experienced by adults with DR-TB , children with this disease also face unique challenges.” 

We also did add a discussion about the disruptions to schooling and other developmentally necessary activities. Although this was not mentioned specifically by our participants, we are aware of the work done to have school programs based in some of the DR-TB treatment wards in South Africa. We thank the reviewer for offering access to these unpublished theses and will reach out to read them for our learning benefit. For the purposes of this paper, however, we will not be able to cite them as they are unpublished and, again, this was not a topic mentioned by our participants. However, we did cite some published literature on the topic of TB and school disruption from China. The discussion section now reads (lines 587-591):

“They result from the outdated models of treatment that unfortunately seem to characterize pediatric DR-TB practices, including prolonged hospitalization (which may disrupt important developmental activities, such as schooling ), reliance on adult formulations of the second-line drugs, coercive adherence strategies, and fear-based infection control practices that lead to prolonged social exclusion for children diagnosed with the disease .”

We also added a note on schooling to Table 2

Reviewer #2: The title is very attractive. The analysis and interpretation do not go in sync.

It is unclear how the respondents categorized as children be 0-14yrs. 

Thank you for this comment. We have clarified that the reason for defining children in this way is because the pediatric population for TB services is defined this way both globally and in South Africa. We have added a sentence that states this in the methods which now reads (lines 112-115):

“Of note, children were defined as those between 0 and 14 years because this is how the pediatric population has historically been defined in the provision of TB services by both the World Health Organization and the South African National Department of Health.”

How did the team even think of eliciting information through FGDs from this group. There is no mention of how many within each group and it is unclear how babies- or even those older could participate in FGDs.

We apologize for the lack of clarity in our methods. We only included children themselves over the age of 6 years, but we included adult caregivers of children of all ages. We have clarified this in the methods as well as noting the number of persons per FGD. The methods now read (lines 143-150):

“A total of 16 children between the ages 7 and 14 years participated in 5 FGDs (with between 3 and 4 children per group), 30 caregivers of children ages 0 to 14 years participated in 7 FGDs (with an between 4 and 5 adults per group) , and 12 providers participated in 3 FGDs (with 4 providers per group). All focus groups contained the same category of participants (i.e., caregiver focus groups contained only caregivers). The caregivers’ and children’s FGD groups were also categorized according to the age groups of the children (0 to 6 years; >6 – 10 years; and >10 to 14 years for children and caregivers, although no children between the ages of 0 to 7 years themselves participated in the discussions due to their ages making participation difficult). “

We also added this as a limitation in the discussion section, which now states (lines 629-630):

“Very young children (i.e., those below the age of 7 years) were also not able to be included in the FGDs.”

The author highlights the caregivers and hardly any information from providers.

It would be better for the authors to confine to care givers and their experiences in dealing with the pediatric group and their challenges befitting the title

The other groups need not be covered.

The recommendations could be for the Health care providers and policy makers.

We have edited the paper as the reviewer suggested and noted that most of the data are from caregivers and children (see lines 178-179, which now read: “A thematic network analysis was performed on the study interviews and transcripts , , with a focus on the experiences of the children and their caregivers.” We did, however, retain certain quotes from participants who were health care providers if these validated the data being reported by children and/or their caregivers.

Very vague analysis and while the recommendations and suggestions are good this does not come from the analysis of this nature

We are sorry that the reviewer feels our recommendations did not come from the analysis. We have tried to update the conclusion to show where the recommendations are linked to the analysis, and we hope the provision of Table 2 in the text can also show how the analysis led to the recommendations. 

Please revise and resubmit

Reviewer #3: The authors identify a significant gap in the literature. This is an important area to study because of the potential for direct impact on care of children with DR-TB in certain SA care contexts and more broadly in terms of awareness-making around a lack of paediatric treatment formulations. This is a strong piece of research, but my main concern is a lack of theoretical basis for conceptualising trauma. The authors take it for granted that the reader knows which conception(s) of trauma are being used for analysis of the primary interview data. It is essential that it be made clear (1) what is meant by trauma; (2) clarify the different figurations of trauma with reference to theory as these figurations structure the findings; (3) place the study and its findings within a clearer theoretical literature - if trauma is the focus, the background literature should reflect this.

We have removed the reference to trauma here as also suggested by reviewer one, except to talk about the framework developed by Das and colleagues which informed our interview guide and early development. We instead emphasize the socio-ecologic model as informing our work as well. Please see lines 164-178, which read:

“Data analysis was based in grounded theory, which centers the analysis on the accounts of the study participants, as opposed to using an already-existing analytic framework .Grounded theory was selected as little is known about the experiences of children who are undergoing treatment for DR-TB. The analysis was, however, informed by a trauma framework developed by Das and colleagues in their study on the illness experiences of adolescents with DR-TB living in Mumbai, India4. “Trauma” is a term often used to describe the experiences of children with chronic and/or life-threatening illness . Trauma may be defined “as physical and psychological experiences that are distressing, emotionally painful, and stressful and can result from an event, series of events, or set of circumstances such as a natural disaster, physical or sexual abuse, or chronic adversity (e.g., discrimination, racism, oppression, poverty).” While the diagnosis of DR-TB and subsequent hospitalization (and thus removal from family and familiar surroundings, often for months at a time) could certainly fit this definition, we utilize instead the term “stressful experiences” throughout this paper. Stress is defined as an experience that is perceived as a threat to wellbeing . The data analysis also utilized a social-ecological model which considers the needs and roles of different actors within the larger societies and communities in which they live.”

METHODS

- It would be good to have more detail on what constituted a 'caregiver' and a 'provider' in the context of the study and how this informed selection.

At the suggestion of reviewer 2, we have removed most reference to providers. However, we have added to the methods section the following information (lines 106-112):

“This was a qualitative study done using focus group discussions (FGDs) among three different groups of participants: 1) health care providers involved in the care of children being treated for DR-TB (including physicians, nurses, and pharmacists) but who were not responsible for the care of the children outside of the health care setting—herein referred to as providers; 2) household caregivers of children being treated for DR-TB who were responsible for looking after the child in his/her household and who could be parents, other relatives, or legal guardians—herein referred to as caregivers.”

- Line 141: typo/syntactical issue

We have corrected this.

- Were transcripts transcribed by researchers or externally? Important to know because of implications for data handling and ethics.

Data were transcribed by external persons who routinely provide such transcription services and are trained and certified in ethical research practice and the ethical handling of sensitive materials. We have added this information to the paper and lines 161-163 now read:

“The recordings were then transcribed (by persons external to the research team but who are trained and certified in ethical research practice as well as the ethical handling of sensitive information)”

- How was the study informed by Das and Colleagues framework? Brief explanation and justification (1 sentence).

We have added the following to the text, and it now reads (lines 170-174):

“The analysis was, however, informed by a trauma framework developed by Das and colleagues in their study on the illness experiences of adolescents with DR-TB living in Mumbai, India4 which described the treatment experiences these adolescents had as painful, damaging to their physical wellbeing and sense of self, and disrupting normal social roles and development activities”.

- Reflexivity: how would the authors' positions as clinicians affect interpretation? How were reflexivity issues incorporated into findings/discussion (why relevant)?

We have added this information on to the statement on reflexivity, which now reads (lines 202-203):

“We note that several of us are engaged in providing care to children with DR-TB as medical

providers and this may have impacted our understanding, analysis, and description of the experiences of the children and caregivers who participated in this study. We may have overly focused on biomedical aspects of their experiences.” We have also addressed reflexivity in the limitations discussion, which now reads (lines 633-636):

“Our practice of reflexivity leads us to acknowledge another limitation of this study in that our work as TB providers (some authors are practicing clinicians as well as researchers) may have led to a focus on biomedical aspects of stressful experiences, both in the analysis of the data and in our conclusions/recommendations.”

BACKGROUND

- Excellent, justified the need for this research.

Thank you for this comment

ETHICS

- How was consent gained for participants younger than 12? Did their guardians consent?

We apologize for the lack of clarity on this. The guardians did consent for participants younger than 12, as noted in lines 106-108 which now read:

“Written consent was obtained from all the participants over the age of 18 years (for caregivers and children), and all children over the age of 12 years also provided formal oral assent to participate in the study. For children under the age of 12 years, only caregiver consent was required. “

FINDINGS

- Needs a working definition/theoretical basis for what constitutes trauma.

We have added to the methods and largely removed the trauma framework from this paper as described earlier.

- 320-322: Provider Participant D does not respond directly to question. Does this reflect the actual transcript? If so, it might be worth mentioning that the response wasn’t direct.

This does reflect the actual transcript, so we have noted that the response was not direct. 

6. PLOS authors have the option to publish the peer review history of their article (what does this mean?). If published, this will include your full peer review and any attached files.

Do you want your identity to be public for this peer review? For information about this choice, including consent withdrawal, please see our Privacy Policy.

Reviewer #1: No

Reviewer #2: No

Reviewer #3: No

---

## [Decision Letter · Decision Letter 1]

4 Sep 2022

Title : “I would watch her with awe as she swallowed the first handful”: a qualitative study of pediatric multidrug-resistant tuberculosis experiences in Durban, South Africa

PONE-D-22-03236R1

Dear Dr. Furin,

We’re pleased to inform you that your manuscript has been judged scientifically suitable for publication and will be formally accepted for publication once it meets all outstanding technical requirements.

Kind regards,

Tai-Heng Chen, M.D.

Academic Editor

PLOS ONE

Reviewers' comments:

Reviewer's Responses to Questions

**Comments to the Author**

1. If the authors have adequately addressed your comments raised in a previous round of review and you feel that this manuscript is now acceptable for publication, you may indicate that here to bypass the “Comments to the Author” section, enter your conflict of interest statement in the “Confidential to Editor” section, and submit your "Accept" recommendation.

Reviewer #1: All comments have been addressed

Reviewer #3: All comments have been addressed

2. Is the manuscript technically sound, and do the data support the conclusions?

Reviewer #1: Yes

Reviewer #3: Yes

3. Has the statistical analysis been performed appropriately and rigorously? 

Reviewer #1: N/A

Reviewer #3: N/A

4. Have the authors made all data underlying the findings in their manuscript fully available?

Reviewer #1: Yes

Reviewer #3: Yes

5. Is the manuscript presented in an intelligible fashion and written in standard English?

Reviewer #1: Yes

Reviewer #3: Yes

6. Review Comments to the Author

Reviewer #1: (No Response)

Reviewer #3: The authors have responded carefully to each of the reviewer comments. I think a greater level of integration between the findings/data and theory (Das) could have been achieved, but acknowledge that this would have been a major rather than minor revision.

7. PLOS authors have the option to publish the peer review history of their article (what does this mean?). If published, this will include your full peer review and any attached files.

Reviewer #1: No

Reviewer #3: No

---

## [Editor Report · Acceptance letter]

9 Sep 2022

PONE-D-22-03236R1 

“I would watch her with awe as she swallowed the first handful”: a qualitative study of pediatric multidrug-resistant tuberculosis experiences in Durban, South Africa 

Dear Dr. Furin:

I'm pleased to inform you that your manuscript has been deemed suitable for publication in PLOS ONE. Congratulations! Your manuscript is now with our production department. 

Kind regards, 

on behalf of

Dr. Tai-Heng Chen 

Academic Editor

PLOS ONE